# Adenosine and the Cardiovascular System: The Good and the Bad

**DOI:** 10.3390/jcm9051366

**Published:** 2020-05-06

**Authors:** Régis Guieu, Jean-Claude Deharo, Baptiste Maille, Lia Crotti, Ermino Torresani, Michele Brignole, Gianfranco Parati

**Affiliations:** 1C2VN INSERM, INRAE, Aix Marseille University, 13005 Marseille, France; jean-claude.deharo@ap-hm.fr (J.-C.D.); baptiste.maille@ap-hm.fr (B.M.); 2Laboratory of Biochemistry, Timone Hospital, 13005 Marseille, France; 3Department of Cardiology, Hôpital La Timone Adultes, 13005 Marseille, France; 4Department of Medicine and Surgery, University of Milano Bicocca, 20122 Milan, Italy; l.crotti@auxologico.it (L.C.); gianfranco.parati@unimib.it (G.P.); 5IRCCS, Department of Cardiovascular, Istituto Auxologico Italiano, Neural and Metabolic Sciences, San Luca Hospital, 20122 Milan, Italy; 6IRCCS, Istituto Auxologico Italiano, Cardiomyopathy Center and Center for Cardiac Arrhythmias of Genetic Origin, 20122 Milan, Italy; 7IRCCS, Laboratory of Cardiovascular Genetics, Istituto Auxologico Italiano, 20122 Milan, Italy; 8IRCCS Istituto Auxologico Italiano, Faint & Fall Programme, Ospedale San Luca, 20122 Milano, Italy; e.torresani@auxologico.it (E.T.); mbrignole@outlook.it (M.B.); 9Department of Cardiology, Arrhythmology Centre and Syncope Unit, Ospedali del Tigullio 16033, Lavagna, Italy

**Keywords:** adenosine receptors, cardiovascular diseases

## Abstract

Adenosine is a nucleoside that impacts the cardiovascular system via the activation of its membrane receptors, named A_1_R, A_2A_R, A_2B_R and A_3_R. Adenosine is released during hypoxia, ischemia, beta-adrenergic stimulation or inflammation and impacts heart rhythm and produces strong vasodilation in the systemic, coronary or pulmonary vascular system. This review summarizes the main role of adenosine on the cardiovascular system in several diseases and conditions. Adenosine release participates directly in the pathophysiology of atrial fibrillation and neurohumoral syncope. Adenosine has a key role in the adaptive response in pulmonary hypertension and heart failure, with the most relevant effects being slowing of heart rhythm, coronary vasodilation and decreasing blood pressure. In other conditions, such as altitude or apnea-induced hypoxia, obstructive sleep apnea, or systemic hypertension, the adenosinergic system activation appears in a context of an adaptive response. Due to its short half-life, adenosine allows very rapid adaptation of the cardiovascular system. Finally, the effects of adenosine on the cardiovascular system are sometimes beneficial and other times harmful. Future research should aim to develop modulating agents of adenosine receptors to slow down or conversely amplify the adenosinergic response according to the occurrence of different pathologic conditions.

## 1. Introduction

Adenosine is a ubiquitous nucleoside that comes from the dephosphorylation of ATP and AMP. It is released specifically during hypoxia, ischemia, inflammation and beta-adrenergic stimulation [1,2,3,4,5]. Adenosine acts on a number of tissues (including the immune and nervous systems) through the activation of four G-coupled membrane receptors, named A1R, A2AR, A2BR and A3R, as a function of their pharmacological properties and primary sequence [6,7,8]. Adenosine also strongly impacts the cardiovascular system mainly through the activation of its receptors. The main effects of adenosine on the cardiovascular system involve heart rate, vasomodulation and blood pressure regulation. The goal of this review is to summarize the impact of adenosine and its receptor activation during several cardiovascular diseases and conditions.

## 2. Source and Mechanism of Action of Adenosine

Adenosine is synthetized in most cells, but the main sources of adenosine in blood are endothelial and muscle cells, through the dephosphorylation of AMP via specific nucleotidases. Adenosine release also occurs after adrenergic stimulation. Part of adenosine production comes from the methionine cycle (see Figure 1). At the extracellular level, adenosine comes from the dephosphorylation of ATP and AMP via the membrane clusters CD39 and CD73, respectively.

Aside of CD39, pyrophosphatases (ENPP1/3) is expressed in many tissues including macrophages and can degrade ATP to AMP leading to enhance adenosine production [9].

Intracellular adenosine leaves the cells via an equilibrative facilitated diffusion system (ENT for equilibrative nucleoside transporter) [10,11]. In the extracellular spaces, the half-life of adenosine is short due to its uptake by red blood cells (see Figure 1). During hypoxia, ischemia, or inflammation, the release of adenylyl nucleotides increases, and adenosine concentration increases at both intra- and extracellular levels [2,4,12].

Schematic representation of adenosine metabolism.

Adenosine is synthesized in most mammalian cells via the dephosphorylation of AMP through nucleotidases. Part of the adenosine comes from the metabolism of methionine. Adenosine is released in the extracellular spaces via an equilibrative nucleoside transporter (ENT). The trigger of adenosine release is mainly hypoxia and inflammation. Adenosine is also converted into inosine and then to xanthine and finally to uric acid, the final product, via adenosine deaminase (ADA) and xanthine oxidase (XO), respectively. In the extracellular spaces, adenosine is formed by the dephosphorylation of ATP and 5′AMP via CD39 and CD73, respectively. Adenosine activates four G-coupled membrane receptors, named A_1_R, A_2A_R, A_2B_R, and A_3_R. Schematically, activation of A_1_R leads to slowing of the heart rate, while activation of A_2_R leads to vasodilation. Finally, A_3_R is implicated in the protection against the ischemia/reperfusion process.

## 3. Adenosine Receptors

Adenosine impacts the cardiovascular system via A_1_, A_2A_, A_2B_, and A_3_ receptor subtypes. All four receptor subtypes have been detected in the heart, with subtype distributions varying from one tissue to another [13]. A_1_R possesses high affinity for adenosine and is expressed throughout the cardiovascular system at high levels in the atria [14]. A_1_R expression varies in cardiac tissues with higher levels in the right atrium than in the left atrium and lower expression in ventricular myocytes than in the atrium [15]. A_1_R is also expressed in smooth muscles and endothelial coronary tissues [16]. A_2A_R is widely expressed in the cardiovascular system but particularly in vessels [17], atria, and ventricular tissues [14,18]. In ventricular myocytes, activation of A_2A_R leads to inotropic properties [19,20]. A_2B_R possesses the lowest affinity for adenosine. A_2B_R is expressed on myocytes and fibroblasts and is reported to modulate ventricular function in animals [20]. A_2B_R is also expressed in smooth muscles of coronary arteries mediating vasodilation [21]. A_3_R myocardial expression is very low. Its expression, however, can be observed within the heart and appears to play a role in coronary artery muscle cells but also in other smooth muscle cells [22,23,24].

A_1_R stimulation leads to a decrease in cAMP production in target cells, which results in the inhibition of protein kinase A (PKA) and voltage-gated calcium channels and activation of phospholipase C [6,7,8]. Activation of A_1_R also leads to direct activation (cAMP-independent) of the inwardly rectifying K+ current (IK_Ado_). Activation of A_2A_R also inhibits voltage-gated Ca++ channels [25]. Schematically, activation of A_1_R leads to bradycardia or atrioventricular block (AVB), while activation of A_2A_R and, to a less extent, of A_2B_R leads to vasodilation via NO and K_ATP_ channels [26]. A_2A_R also inhibits L Type calcium currents [25]. Despite opposite effects on cAMP production in target cells, there is some overlap between the cardiovascular effects following A_1_R or A_2A_R activation. As an example, activation of A_1_R leads mainly to a decrease in heart rate, but KO mice for A_2A_R exhibit tachycardia, suggesting that A_2A_R activation leads to a reduction in heart rate [27]. A_3_Rs are implicated more specifically in ischemia/reperfusion protection, but all receptor subtypes seem to be implicated in ischemia myocardium protection [7]. Finally, A_2A_ and A_2B_ receptors are expressed on platelets, where their activation leads to antiplatelet properties via calcium flux inhibition [28].

Finally, a specific pharmacological profile, called the receptor of reserve, was described in some diseases, such as coronary artery disease or syncope [29]. It seems that this type of receptor, which is characterized by maximal biological effects (evaluated by cAMP production) while only a weak proportion of receptors are activated by the ligand [30], is an adaptive response to compensate for low adenosine levels, low receptor expression levels or both [29]. The precise role of this kind of receptor needs further investigation.

## 4. Effects of Adenosine on Vessels

Adenosine is known to regulate coronary blood flow (CBF) [31] and exerts potent vasodilatory effects in most vascular beds of mammalian species [32]. These effects are secondary to the activation of A_2A_R [17,30,33] and A_2B_ receptors [34] and occur via the production of cAMP in smooth muscle cells [17] and the activation of Kv, K_ATP_ channels and NO pathways, both in peripheral arterial vessels [35,36] and in coronary arteries [26,37]. Mice with A_2A_R KO exhibit hypertension, tachycardia, and platelet aggregation abnormalities [27].

## 5. Effects of Adenosine on the Sinus Node and Atrio–Ventricular Junction (Figure 2)

Adenosine suppresses the activity of cardiac pacemakers at the sinus node, atrioventricular node and His bundle [38]. Adenosine has negative dromotropic effects that result in an increase in the PR interval as well as in complete AVB. This effect is secondary to the hyperpolarization of cell membranes on the AVN [39]. This action is mediated mainly through the A_1_R (see Figure 2) via the direct (cAMP independent) activation of the inwardly rectifying IK_Ado,Ach_ currents [40]. Adenosine and acetylcholine, through A_1_R and M_2_ muscarinic receptor activation, respectively, have very similar effects on cardiomyocytes, both leading to the activation of IK_Ado,Ach_. Adenosine through A_1_R induces the inhibition of hyperpolarization-activated (funny) currents [25,41]. Funny channels are activated by cAMP as well as by voltage hyperpolarization and are implicated in the increase in heart rate by catecholamines [42]. Thus, these antiadrenergic effects can also play a role in the negative chronotropic effect of adenosine [39]. The negative chronotropic action of adenosine on the myocardium was also attributed to the inhibition of the inward calcium current (Ica) [40].

Adenosine shortens action potential duration and increases refractoriness [39,40,43]. These effects are mainly mediated by the activation of A_1_R. In the atria, adenosine exerts direct and indirect anti-beta-adrenergic effects. The activation of IK_Ado, Ach_ via A_1_R leads to shortening of action potential duration and refractoriness [43], thus facilitating reentry mechanisms and atrial arrhythmias [39]. Thus, overexpression of A_1_R is associated with bradycardia, delayed conduction through the sinoatrial and atrioventricular nodes, atrial arrhythmia, and ventricular hypertrophy [44,45].

It was also supposed that the activation of A_2A_R stimulates the ryanodine receptors that control part of the intracellular calcium flux from the sarcoplasmic storage site [46].

On the sinus node, the activation of vagal efferent fibers via the modulation of M2 R and via the activation of A_1_R have very similar effects, leading to a slower heart rate.

AVN: atrioventricular node

GIRK: G-protein inwardly rectifying potassium channels

K: _ATP_, K_V_: ATP-sensitive and voltage-sensitive potassium channels, respectively.

M2: M2 muscarinic receptors

SN: sinus node

VDCC: voltage-dependent calcium channels

## 6. Adenosine and Ventricular Myocytes

Adenosine exerts anti-beta-adrenergic effects in ventricular myocytes and reduces the cAMP concentration via A_1_R activation [47]. Adenosine inhibits the adrenergic-dependent increase in inward L-Type Ca++ currents and reduces the amplitude of delayed afterdepolarization [48]. Adenosine terminates episodes of ventricular tachycardia and abolishes the delayed after depolarization associated with digoxin toxicity [49]. These antiarrhythmic effects of adenosine at the ventricular level are mediated mainly by its anti-beta-adrenergic effects [50].

## 7. Clinical Aspects

### 7.1. Adenosine and Atrial Fibrillation

Atrial fibrillation (AF) is the most common arrhythmia that affects 1% to 4% of the population [51]. The onset of AF is associated with both vagal and sympathetic activation, with vagal activity being responsible for the decrease in refractory period [52], while sympathetic activation is responsible for atrial activation via calcium release [53]. Both intrinsic and extrinsic autonomic nervous system influences are implicated in the onset of AF [54]. Since the onset of atrial fibrillation is mostly due to ectopic beats originating in the pulmonary veins [55,56], isolation of the pulmonary vein by catheter ablation is effective [57,58]. The adenosine test has a high predictive value for AF recurrence after pulmonary vein isolation [59].

Several experimental and observational findings suggest that the adenosinergic system is likely to be implicated in the onset of paroxysmal AF and probably in its maintenance in persistent form. The administration of exogenous adenosine induces AF in susceptible patients [60,61]. In patients suffering from supraventricular tachycardia, adenosine administration led to AF or flutter in 12% of cases, and atrial premature complexes occurred in 58% of cases [62]. In patients with paroxysmal AF, the infusion of adenosine and isoproterenol in sinus rhythm was able to induce atrial ectopic beats in most patients [63]. High adenosine plasma levels (APL) have been found in the left atria of patients during episodes of paroxysmal and persistent AF, APL normalized during spontaneous resolution of this arrhythmia or after direct cardioversion treatment [64] (see Table 1). Peripheral APL is also high in permanent AF, and this finding has been attributed to peripheral hypoxemia caused by the decrease in left ventricular output [64]. High serum uric acid, the final product of adenosine metabolism, was found to be associated with AF [65,66,67]. A_2A_ receptors are expressed in the right atrium in humans, and their distribution overlaps ryanodine receptors [46]. Abnormal expression levels of A_2A_R, the activation of which leads to calcium release from the sarcoplasmic reticulum of cardiomyocytes, have been described in patients with AF [68]. Finally, heterogeneous expression of A_1_R and high expression of G-protein-coupled inwardly rectifying potassium channels have been observed in the right atria of susceptible patients, suggesting that adenosine-induced AF is driven by localized reentry in the right atria [69]. Thus, both A_1_R and A_2A_R are suspected to be implicated in AF pathophysiology.

Adenosine may initiate AF through three main mechanisms: (i) Sympatho-excitatory effects; (ii) shortening the refractory period via the activation of the inward rectifying K+ current (IK_ado_) [42]; and (iii) direct stimulation of the pulmonary vein tissue [61]. This last effect seems to be sufficient to trigger AF [61].

### 7.2. Effects of Exogenous Adenosine

Adenosine or ATP are commonly used to treat supraventricular tachycardia [89]. The side effects of exogenous adenosine include bronchospasm [90], chest pain [91], bradycardia, and sometimes complete AVB [92]. This effect may be used for the diagnosis of unexplained syncope (see below). Sometimes, the administration of exogenous adenosine has resulted in asystole followed by myoclonic jerk or death [93,94]. It seems that exogenous adenosine may cross the feto–placental barrier since fetal bradycardia was observed after adenosine administration for maternal supraventricular tachycardia [95]. Ventricular tachycardia or torsade de pointes have rarely been observed after adenosine administration [96,97,98]. By provoking transient bradycardia followed by sinus tachycardia, adenosine was shown in a single study on 18 cases and 20 controls to induce abnormal QT responses in patients with long QT syndrome (LQTS) compared to controls, suggesting that adenosine administration might be useful for distinguishing patients with LQTS from healthy controls [99]. However, this study was never replicated, and the “adenosine challenge test” is not actually considered part of the diagnostic worked out in cases of suspected LQTS. Torsade de pointes have also been observed after adenosine administration in patients with [100,101] or without QT prolongation [98,102]. Most effects of exogenous adenosine administration are summarized in Table 2.

### 7.3. Adenosine and Reflex (Neurohumoral) Syncope

Reflex (neurohumoral) syncopes (NHS) are frequent (1%–3%) in the general population and may severely alter the quality of life of patients. NHS is responsible for 3% to 5% of emergency entrance and 1% to 2% of hospitalizations with a poor outcome in nearly 7% of cases [104,105]. NHS episodes occur at least once during life in 50% of the whole population. NHS is characterized by a partial (presyncope) or complete loss of consciousness due to a decrease in heart rate and systemic blood pressure in the absence of any heart structural abnormalities. Loss of consciousness is often preceded by prodromes, including discomfort, nausea, vomiting, abdominal pain, but sometimes the loss of consciousness occurs at outset, without or with very short prodromes [73,74,103]. Symptoms may be reproduced by the head up tilt test (HUT) [106] or sometimes by ATP or adenosine administration [107].

There is evidence that the adenosinergic system is implicated in the pathophysiology of NHS. Patients with typical vasovagal syncope (VVS) have a high baseline APL that further increases during HUT-induced syncope [70], high expression of A_2A_R [71] and specific single nucleotide polymorphism (CC variant) in the second exon of the gene encoding the A_2A_R [72]. Hyperoxia, which induces a drop in APL, restores hemodynamic status during HUT [108], thus suggesting that adenosine modifies the behavior of heart rate and blood pressure during HUT. In VVS, high APL stimulates low affinity A_2A_R (K_D_ 1.8 µM) that causes vasodilation [30]. The high basal adenosine plasma level may be secondary to the high A_2A_R expression, which could be itself secondary to a genetic predisposition, as documented by a specific polymorphism. Indeed, it was shown that activation of A_2A_R produces an increase in ENT expression favoring the release of adenosine in extracellular spaces [109] and thus an increase in extracellular adenosine, creating a vicious circle (see Figure 3).

More recently, a low adenosine syncope subtype was described that is characterized clinically by the presence of short or no prodromes. In this kind of syncope, the loss of consciousness may be reproduced by ATP or adenosine administration [73,74,103,110].

In low APL syncope, the adenosine concentration is under the affinity (K_D_) value for the activation of A_1_R (K_D_ 0.8 µM) [111]. Thus, a small increase in APL (due to transient hypoxia, beta-adrenergic stimulation, weak decrease in blood pressure or unnoticed inflammatory process) may lead to the activation of A_1_R, creating a sinus arrest or an AV block, mostly via the activation of IK_Ado_ that induces hyperpolarization [25] (see Figure 2). In this case, the APL is sufficient to activate high affinity A_1_R but not low affinity A_2A_R, and thus, vasodilation is less important. In this subtype of syncope, the use of theophylline, a nonspecific adenosine receptor antagonist, is often effective in a preventive manner because theophylline binds to A_1_R and prevents adenosine binding [75,112]. Finally, if high APL plays a role in explaining VVS in patients, the cause of low APL in low adenosine syncope remains unknown. Abnormalities in adenosine metabolism, including ENT or ADA expression, which are currently under investigation, are potential factors.

### 7.4. Adenosine and Hypoxia

#### 7.4.1. Syncope in Hypoxic Conditions

During breath-hold diving, a transient loss of consciousness (TLOC) may occur, which is attributed to cerebral hypoxemia. Adenosine release, induced by hypoxia, may precipitate TLOC. Indeed, high APL was observed in basal conditions and further increased during breath-hold [76,113]. The adenosinergic profile of divers is different from that of patients with VVS since the HUT test was positive in only 16% of divers compared with 80% in patients with VVS [76]. The CC variant of the SNP was observed in the majority (56%) of patients with VVS, whereas the TC variant was found in 56% of divers and 61% of divers who had previous TLOC during diving [76]. Heart rate and O_2_ saturation were both correlated with APL in breath-hold divers. Furthermore, divers with the highest APL at the end of breath-hold were the same ones who reported previous TLOC during diving [113]. In addition, hypercapnia occurs during breath-hold diving. Adenosine mediates the hypercapnic response in the rat carotid body via A_2A_R and A_2B_R, which promotes bradycardia via the carotid sinus pathway [114]. In conclusion, we hypothesize that in some cases, TLOC may be secondary to severe bradycardia, sinus arrest or AVB in the context of high APL.

#### 7.4.2. Obstructive Sleep Apnea Syndrome

Obstructive sleep apnea syndrome (OSAS) is characterized by episodes of partial or complete occlusion of upper airways during sleep, which is associated with hypoxemia. Sleep apnea is a serious condition that has been reported to affect more than 22 million Americans who suffer from a lack of restorative sleep. Typical symptoms of sleep apnea include heavy snoring, excessive daytime sleepiness or fatigue, difficulty with concentration or memory, and waking during the night feeling short of breath. Untreated sleep apnea can lead to serious health consequences, including chronic hypoxemia, systemic and pulmonary hypertension, and heart failure [115,116]. Hypoxemia occurs during sleep in OSAS. It was shown that adenosine plasma levels and its metabolites, including uric acid (see Figure 1), increase during apnea in patients with OSAS [77,78,79]. In most cells, hypoxia induces the stabilization of HIF (hypoxia-inducible factor) alpha subunit, which inhibits adenosine kinases, leading to the inhibition of rephosphorylation of adenosine in ATP and then favoring the accumulation of adenosine in both intra- and extracellular spaces [117]. The decrease in PaO_2_ and the lack of ATP also explain the increase in blood lactates during apnea [118]. In the brainstem, due to its hyperpolarizing properties on neurons, adenosine participates in the initiation of sleep. During wakefulness, adenosine levels gradually increase in areas of the brain that are important for promoting arousal, in particular the reticular activating system in the brainstem. The extracellular adenosine concentration increases during sleep. Therefore, high levels of adenosine cause sleep [119]. Caffeine, a nonspecific adenosine receptor antagonist, works to inhibit sleep by blocking the action of adenosine within the brain, which facilitates wakefulness. Finally, an A_1_R agonist (N (6)-p-sulfophenyladenosine) suppresses apnea during all sleep stages in rats, and this effect occurs via the peripheral nervous system [120]. In summary, adenosine during sleep apnea is released in response to hypoxia and may promote sleep but inhibits apnea episodes through A_1_R activation.

#### 7.4.3. Altitude Hypoxia

Physiological adaptation to hypoxia is a source of intensive investigations. In daily life, hypoxia may be due to reduced barometric pressure, as in altitude-induced hypoxia conditions, while in pathophysiological conditions, hypoxia may be secondary to pulmonary obstructive or restrictive disease. Hypoxia may also be acute or chronic depending on the cause of hypoxia and the features of the responsible disease. Unlike sleep apnea or breath-holding, altitude hypoxia is not associated with hypercapnia and acidosis. Thus, hypercapnia-induced A_2_R stimulation in carotid glomus cells lacking at altitude. In all cases, however, the extracellular accumulation of adenosine is a crucial protective step to limit cellular damage during hypoxia [1,121]. Hypoxia leads to an increase in adenosine extracellular concentration [1], which favors vasodilation through the activation of adenosine A_2A_ and A_2B_ receptors [122,123]. Short exposure (3 h) to low PO_2_ increases the mRNA and protein of the A_2B_ receptor subtype independently of the HIF pathway [123]; chemical hypoxia also upregulates A_2A_R expression via the NF-kappa B pathway [124]. It is unclear why acute altitude induces simultaneous pulmonary vasoconstriction and peripheral vasodilation. The latter, however, after 6–8 h, is overwhelmed by reflex vasoconstriction due to sympathetic activation by chemoreflex stimulation and endothelial dysfunction related to oxidative stress.

The extracellular concentration of adenosine during hypoxia is mainly controlled by ADA [1] and by ENT-1 [10,11,125] with the expression of ENT-1 being mainly controlled by A_2B_R in erythrocytes [126] and HIF in human dermal microvascular endothelium and in endothelium from the human saphenous vein [125]. High altitude exposure is a well-known cause of hypoxia due to reduced partial oxygen pressure in the alveolar air-hypobaria. In these conditions, an increase in the adenosine extracellular level was observed, which was attributed to both an increase in CD73 activity [126], leading to the increase in dephosphorylation of ATP into 5′AMP and a decrease in ENT-1 erythrocyte expression [80], leading to the inhibition of adenosine uptake by erythrocytes (see Figure 1). It was found that an increase in APL via activation of A_2B_R induced downregulation of the ENT1 transporter in erythrocytes, which participates in faster acclimatization to high altitude upon re-ascent [80]. Furthermore, adenosine via A_2B_ induces the production of 2,3-BPG by erythrocytes after only a few hours of hypoxia exposure [126]. 2,3-BPG favors O_2_ liberation by hemoglobin in an allosteric manner. The adenosine pathway also impacts erythrocyte glucose metabolism, favoring more O_2_ delivery during adaptive mechanisms to high altitude-induced hypoxia [127]. Altitude hypoxia also seems to modify platelet reactivity through the purinergic pathway. Indeed, hypobaric hypoxia significantly reduced the ability of a fixed concentration of cangrelor to inhibit ADP-induced aggregation and increased basal vasodilator-stimulated phosphoprotein (VASP) phosphorylation [128]. In summary, the adenosinergic response seems to be a crucial adaptative response to altitude hypoxia.

### 7.5. Myocardial Ischemia Reperfusion Protection

The contribution of adenosine to the regulation of coronary blood flow was first postulated by Berne [31]. Adenosine is known to induce vasodilation, via A_2A_R and A_2B_R [26,30,37], leading to an increase in blood flow and oxygenation. While it was established that adenosine is not required for the regulation of CBF at rest, its contribution is crucial during effort when myocardial oxygen requirements are not sufficiently met or during ischemia process [129,130]. The contribution of A_2A_R and A_2B_R to coronary vasodilation is particularly marked in the adaptive response to ischemia process. Adenosine accumulates during hypoxia or ischemia, due to the imbalance between O_2_ supply and need resulting in an imbalance between ATP synthesis and consumption [131]. Interstitial adenosine concentration increase during ischemia but decrease with repetitive brief ischemia [132,133]. The administration of exogenous adenosine reduced infarct size in an animal model of myocardial ischemia [134].

Cardioprotection seems to involve A_1_ R [135,136,137] and A_3_ R [135,138] through a proteine kinase C, K_ATP_ channels and antiadrenergic effects [139]. Over expression of A_1_ R is associated with increased resistance to ischemia, while the precise mechanism of A_3_R-induced cardioprotection remains elusive [138].

In a clinical point of view, adenosine plasma was found to be high in the coronary sinus of patients with severe coronary artery disease that decreased after percutaneous transluminal angioplasty (PTCA) [140]. The use of dipyridamole that increases extracellular adenosine concentration improved tolerance during exercise stress test [141]. Intra coronary adenosine administration may also be beneficial during PTCA by improving ischemia reperfusion injuries, left ventricular farction ejection and by improving clinical outcome [142,143] and by while its administration improver. Exogenous adenosine is used commonly in patients with coronary artery disease to determine coronary reserve, to induce a maldistribution and to image such maldistribution [144].

Thus the release of endogenous adenosine during ischemia seems to be an adaptive mechanism to improve coronary blood flow.

### 7.6. Adenosine and Vascular Injury and Repair

Adenosine also plays a major role in promoting wound healing and vascular tissues repair [145]. Thus A_2_ adenosine receptor subtypes are required to new matrix production and angiogenesis while A_1_ R up regulates vascular endothelial growth factor from monocytes and contributes to new vessels formation [146]. The activation of adenosine receptors promotes endothelial cell proliferation and stimulates angiogenesis factors production. In this context, activation of A_2A_ R promotes wound healing during experimental tissue injury [147].

Vascular muscle cell proliferation is an important component of vascular remodeling which is regulated partly through A_2B_ receptors [148] and it was shown that activation of A_2B_R protects against vascular injury [149].

Finally drugs that increase adenosine plasma levels, improved endothelial function [150]. Thus targeting A_2A_ and A_2B_ receptors are promising for vascular healing after injury [151,152].

### 7.7. Adenosine and Systemic Hypertension

The adenosinergic system plays a major role in the control of heart rate and blood pressure. Adenosine inhibits alpha-1 adrenoreceptor responses in physiological and pathophysiological conditions [153,154]. A_2A_R and A_2B_R are expressed in myocytes of arteries including large arterial trunks [17], while A_1_R are expressed in the renal microcirculation [155], where they promote renal vasoconstriction, mediate tubuloglomerular feedback [156], and augment renal vasoconstriction induced by angiotensin-II [157,158] and norepinephrine [155]. A_1_R antagonists increase sodium excretion in rats [159], and A_1_R activation leads to vasoconstriction in the aorta and mesenteric arteries [160,161] and sodium reabsorption in the kidney [82,159].

A_2A_R mediates vasodilation in most arteries, including the aorta [17,33], kidney arteries [81], and skeletal muscles [162], and promotes natriuresis via the increase in blood flow in renal medulla vessels [157,163]. A_2B_ mediates vasodilation in most arteries, including the aorta [164], and mesenteric arteries [165]. In the renal medullary circulation, both A_2A_ and A_2B_R regulate blood pressure by increasing blood flow and thus enhancing sodium excretion [166]. However, A_2B_R may promote kidney fibrosis through the production of endothelin 1, thus promoting chronic hypertension [167].

An overexpression of A_2A_R was described on lymphocytes of patients suffering from essential hypertension, which normalized after alpha 1 blockade but not after administration of beta blocking agents [168]. The increase in A_2A_R expression associated with the increase in the affinity of receptors may be a compensatory mechanism aimed at increasing vasodilation to compensate for high blood pressure. Recently, it was shown that adenosine receptors influence hypertension in Dahl salt-sensitive rats and that this influence depends on sex. Indeed, beneficial effects of A_1_R or A_2B_R KO were limited to females [155]. High APL was found in patients with essential hypertension [169], while high A_2B_ receptor expression was found in patients with hypertension associated with fibromuscular dysplasia [169].

A_3_R also participates in the regulation of blood pressure by affecting the steady-state level of cAMP in smooth muscle cells [23].

In summary, due to its impact on vasodilation, heart rate, and sodium excretion regulation, the adenosinergic system strongly impacts blood pressure regulation. In this perspective the use of specific A_2A_R agonists or A_1_R antagonists to lower blood pressure should be promising.

### 7.8. Adenosine and Pulmonary Hypertension

Primary pulmonary hypertension (PAH) is a severe disease associated with a high incidence of death [83,170] and is characterized by elevation in pulmonary artery pressure followed by right ventricular hypertrophy. The pathophysiology of this condition remains unclear, and PAH seems to be of multifactorial origin. Acute altitude exposure is associated with increased pulmonary pressure (involving endothelin 1 receptor activation) [84]. Sometimes, chronic high altitude exposure may also promote pulmonary hypertension [171].

Dysfunction of endothelial and smooth cells seems to play a major role [172]. Dysregulation of the renin–angiotensin–aldosterone system has also been advocated [85]. Experimental data seem to indicate that adenosine may be a potential endogenous regulator of PAH development by regulating smooth muscle proliferation and vasodilator properties. Adenosine via A_2A_R exerts vasodilation in pulmonary circulation since the lack of adenosine A_2A_R confers pulmonary arterial hypertension in mice [86], while in situ adenosine infusion reduced pulmonary vascular resistance in humans [88]. Furthermore, A_2A_R agonist attenuates the progression of pulmonary hypertension in an experimental model of PAH [173]. Endogenous adenosine measured in distal pulmonary arteries was found to be lower in PAH than in controls [174,175]. Thus, the low APL in the pulmonary arteries may participate in the high pulmonary vascular resistance (PVR) in PAH. This fact is supported by the correlation between APL and PVR [174]. Activation of A_2A_R leads to strong vasodilation via pulmonary endothelial NO synthesis, smooth cell hyperpolarization and inhibition of collagen deposition [173]. Thus, targeting A_2A_R could potentially serve as an efficient treatment of PAH, especially since there is, currently, no convincing treatment available.

### 7.9. Adenosine and Heart Failure

Acute heart failure (AHF) and chronic heart failure (CHF) are the most common complications of cardiovascular disease. The prevalence of HF depends on the definition applied, but it is approximately 1%–2% of the adult population in developed countries, rising to ≥10% among people >70 years of age [87].

CHF results in the release of many neurohumoral factors, including catecholamines, renin angiotensin, and cytokines. While a number of cytokines worsen cardiac performance [176], some of them, like adenosine, may have protective effects.

High APL has been measured in both AHF and CHF [87,88,173,174,175,176,177,178,179] and was attributed to cardiac failure-induced hypoxemia and to a downregulation of adenosine deaminase [178,179,180].

A_1_R KO mice exhibit a normal cardiac phenotype in the absence of stress, while overexpression of A_1_R leads to an increase in myocardium protection against ischemia [181]. This effect was attributed to the inhibition of calcium overload in the sarcoplasmic reticulum that occurs during ischemia [182].

A_2A_R gene expression was found to be increased in patients with dilated cardiomyopathy and in patients with terminal CHF who normalized after transplantation [183]. A_2A_R deletion also leads to cytokine release and proinflammatory effects, yet cytokine release participates in the pathophysiology of CHF [184]. A_2A_R activation also leads to inotropic effects [19,20]. Finally, a polymorphism (SNP rs4822489) was found to be associated with the severity of CHF [185].

In AHF, high APL and high A_2A_R expression have been observed [179]. Adenosine release is secondary to tissue hypoxemia, and the activation of A_2A_R may be beneficial by promoting inotropic function. In cardiogenic shock, A_2A_R expression was lower than in AHF. In AHF, the priority may be to restore myocardium function by increasing both APL and A_2A_R expression, while in cardiogenic shock, the priority may be to restore a sufficient blood pressure level by decreasing A_2A_R expression leading to vasoconstriction [179].

Regarding A_2B_R, it was shown that its inhibition reduces left ventricular dysfunction and ventricular arrhythmias after experimental myocardial ischemia [186]. Regarding A_3_R, its deficiency exerts protective effects on pressure-overload-induced left ventricular hypertrophy and dysfunction [187].

In summary, by impacting vascular tone, myocardial inotropy and sodium excretion, the adenosinergic system appears to be an important modulator in heart failure.

## 8. Conclusions and Future Directions

The adenosinergic system appears to have a key role in the adaptive response of the cardiovascular system both in physiological and pathophysiological conditions.

Adenosine release is likely to participate in the pathophysiology of the disease in at least two conditions, namely, atrial fibrillation and some forms of reflex (neurohumoral) syncope. The adenosinergic system appears to have a key role in the adaptive response of the cardiovascular system in pulmonary hypertension and heart failure, the most relevant effects being slowing of heart rhythm, coronary vasodilation, and decreasing blood pressure. Finally, in other cases, such as altitude or apnea-induced hypoxia, obstructive sleep apnea, and systemic hypertension, adenosine is rather a consequence of the disease in the context of an adaptive response.

Due to its metabolism and quick release, adenosine, via the activation of its receptors, allows very rapid adaptation of the cardiovascular system, with opposite effects to those of the catecholaminergic system. Owing to the complex interaction of adenosine with cardiovascular pathophysiology, the effects of adenosine on the cardiovascular system are sometimes beneficial and sometimes harmful. Future research should aim to develop modulating agents of adenosine receptors to slow down or conversely amplify the adenosinergic response according to the occurrence of different pathologic conditions.

## Figures and Tables

**Figure 1 jcm-09-01366-f001:**
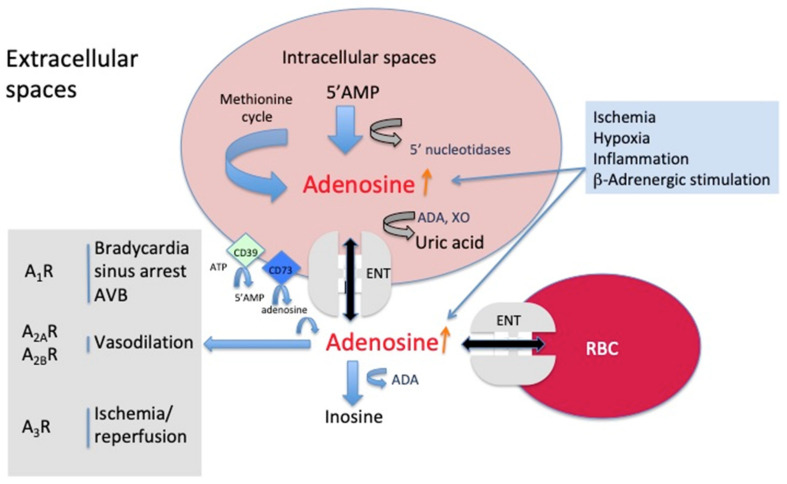
Representation of adenosine metabolism.

**Figure 2 jcm-09-01366-f002:**
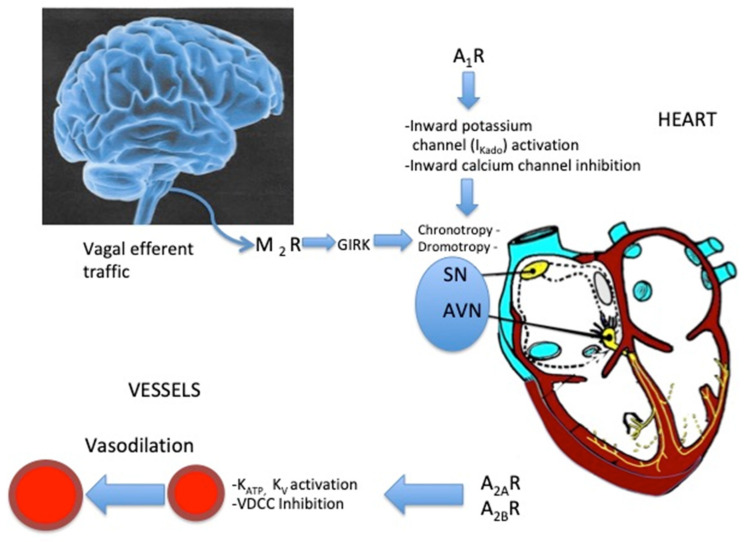
Schematic representation of the action of adenosine or vagal efferent fibers on the heart and vessels.

**Figure 3 jcm-09-01366-f003:**
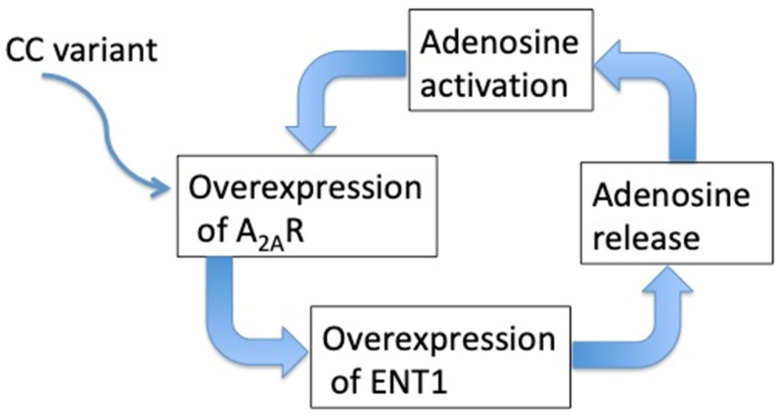
the vicious circle of adenosine release in vasovagal syncope.

**Table 1 jcm-09-01366-t001:** Main purine abnormalities in human cardiovascular disease.

Disease	Purinergic Abnormalities
**Atrial fibrillation**	High APL in the left atria [64].
	High A_2A_R expression in left atria [46,68]. Heterogenous expression of A_1_R in right atria [69]
**Reflex (neurohumoral) syncope**	

Vasovagal syncope	
	High APL [70]
	High A_2A_R expression [71,72]
	CC variant in the second exon of the gene [72]
Syncope without prodrome, normal heart and normal electrocardiogram	
	Low APL [73,74]
	Low A_2A_R expression [73]
**Hypoxic conditions**	
**TLOC during dive**	High APL [75,76]
*Sleep apnea*	High hypoxanthine and xanthine serum levels [77]
	High APL and serum uric acid [78,79]
*Altitude hypoxia*	High APL, high A_2B_R expression and low ENT1 [80]
**Hypertension conditions**	
*Systemic hypertension*	High APL [81]
	High A_2A_R expression [82]
*Pulmonary Hypertension*	Low APL in pulmonary arteries [83,84]
**Heart failure**	CHF: High APL [85,86]
	Increase in A_2A_R expression [87]
	AHF: High APL and high A_2A_R expression [88]

APL: adenosine plasma level; A_1_R: adenosine A_1_ receptor; A_2A_R: adenosine A_2A_ receptor; A_2B_R: adenosine A_2B_ receptor; AHF: acute heart failure. CHF: chronic heart failure. TLOC: transient loss of consciousness.

**Table 2 jcm-09-01366-t002:** Main effects on heart rhythm after exogenous adenosine administration.

**Supra Ventricular Level**	Interruption of tachycardia [89]
	AVB [92]
	AVB in low adenosine syncope patients [73,103]
	Asystole, myoclonic jerk [93]
	Fetal bradycardia [95]
	Atrial fibrillation [60]
	Atrial premature complex [62]
	Atrial ectopic beats [63]
	Flutter [62]
**Ventricular Level**	Increased QT interval in LQTS [99]
	Torsade de pointes in LQTS [100]
	Ventricular tachycardia [96,97]
	Torsade de pointes [98,102]

AVB: atrioventricular block; LQTS: long QT syndrome.

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
