# Peer review of "Adenosine and the Cardiovascular System: The Good and the Bad"

_jcm, 2020, doi:10.3390/jcm9051366_

Round 1
Reviewer 1 Report
This review article introduces the influence of adenosine on the cardiovascular system, particularly on some common cardiovascular diseases and conditions, such as hypertension, heart failure, and arrhythmias. Given a short half-life of the adenosine and its opposite action through affecting different receptors, including A1R, A2AR, A2BR, and A3R, it would be more interesting if this article could make a perspective about the application of the selective agonist or antagonist of adenosine receptor to treat cardiovascular diseases. With this respect, for instance, a specific A2AR agonist or A1R antagonist to lower blood pressure in hypertension.
Author Response
Dear Sir
As suggested, the sentence" In this perspective the use of specific A2A R agents or A1R antagonists to lower blood pressure should be promising"
has been added at the end of 7.7 paragraph
Reviewer 2 Report
The authors have done a good job reviewing adenosine in the cardiovascular system.
One aspect which is missing and should be highlighted in at least a few paragraphs would be the impact of adenosine on regulation of vascular healing and the post arterial repair.
There is a broad literature on targeting A2A and A2B for vascular healing after injury.
Author Response
Dear Sir
A new paragraph has been added. In this context, 8 new references (137-144) have been added.
7.6 Adenosine and vascular injury and repair
Adenosine also plays a major role in promoting wound healing and vascular tissues repair [137]. Thus A2adenosine receptor subtypes are required to new matrix production and angiogenesis while A1 R up regulates vascular endothelial growth factor from monocytes and contributes to new vessels formation [138]. The activation of adenosine receptors promotes endothelial cell proliferation and stimulates angiogenesis factors production. In this context, activation of A2A R promotes wound healing during experimental tissue injury [139].
Vascular muscle cell proliferation is an important component of vascular remodeling which is regulated partly through A2B receptors [140] and it was shown that activation of A2BR protects against vascular injury [141].
Finally drugs that increase adenosine plasma levels, improved endothelial function [142]. Thus targeting A2A and A2B receptors are promising for vascular healing after injury [143-144].